# Carvacrol Treatment Reduces Decay and Maintains the Postharvest Quality of Red Grape Fruits (*Vitis vinifera* L.) Inoculated with *Alternaria alternata*

**DOI:** 10.3390/foods12234305

**Published:** 2023-11-29

**Authors:** Hongying Li, Jie Ding, Chunyan Liu, Peng Huang, Yifan Yang, Zilu Jin, Wen Qin

**Affiliations:** 1College of Food Science, Sichuan Agricultural University, Ya’an 625014, China; lihying1998@163.com (H.L.); 2016106005@yibinu.edu.cn (P.H.); 18599994741@163.com (Y.Y.); 15697053681@163.com (Z.J.); 2College of Food Science, Sichuan Tourism University, Chengdu 610100, China; dingjiedream@163.com; 3Chengdu Kuafu Technology Co., Ltd., Chengdu 610100, China; chunyan202103@163.com; 4Department of Quality Management and Inspection and Detection, Yibin University, Yibin 644000, China

**Keywords:** *Alternaria alternata*, isolation and identification, biological characteristics, natural bacteriostatic agent, postharvest disease, antifungal activity

## Abstract

In this study, we isolated and identified pathogenic fungi from the naturally occurring fruits of red grapes, studied their biological characteristics, screened fifteen essential oil components to find the best natural antibacterial agent with the strongest inhibitory effect, and then compared the incidence of postharvest diseases and storage potential of red grapes treated with two concentrations (0.5 *EC_50_*/*EC_50_*) of essential oil components (inoculated with pathogenic fungi) during storage for 12 d at room temperature. In our research, *Alternaria alternata* was the primary pathogenic fungus of red grapes. Specifically, red grapes became infected which caused diseases, regardless of whether they were inoculated with *Alternaria alternata* in an injured or uninjured state. Our findings demonstrated that the following conditions were ideal for *Alternaria alternata* mycelial development and spore germination: BSA medium, D-maltose, ammonium nitrate, 28 °C, pH 6, and exposure to light. For the best *Alternaria alternata* spore production, OA medium, mannitol, urea, 34 °C, pH 9, and dark conditions were advised. Furthermore, with an *EC_50_* value of 36.71 μg/mL, carvacrol demonstrated the highest inhibitory impact on *Alternaria alternata* among the 15 components of essential oils. In the meantime, treatment with *EC_50_* concentration of carvacrol was found to be more effective than 0.5 *EC_50_* concentration for controlling *Alternaria alternata*-induced decay disease of red grapes. The fruits exhibited remarkable improvements in the activity of defense-related enzymes, preservation of the greatest hardness and total soluble solids content, reduction in membrane lipid peroxidation in the peel, and preservation of the structural integrity of peel cells. Consequently, carvacrol was able to prevent the *Alternaria alternata* infestation disease that affects red grapes, and its *EC_50_* concentration produced the greatest outcomes.

## 1. Introduction

*Alternaria Nees*, a member of the *Pleosporaceae* family in the *Dothideomycetes* class of *Ascomycota*, is a highly prevalent fungus that includes saprophytic, parasitic, and endophytic strains. Among these, *Alternaria alternata* (*A. alternata*) is a saprophytic fungus capable of surviving in various substrates, including water, soil, and air. Additionally, certain strains have the ability to re-colonize and grow on host tissues affected by diseases, exacerbating the severity of the infection [1]. *A. alternata*, as a parasitic fungus, has the ability to extensively infect fruits and vegetables during field cultivation, storage, and through direct contact. Significant losses in the nutritional and commercial value of crops occur as a result of this pervasive contamination, both in the field and during the postharvest procedures [2]. Currently, over 95% of reported cases indicate that *Alternaria* has the ability to facultatively parasitize various crops. It is also recognized as a major cause of diseases of blueberries, cherries, maize, tomatoes, and other fruits and vegetables [3,4,5]. In addition, *Alternaria* is known to produce mycotoxins that can be harmful to humans and animals. Some of these mycotoxins include alternariol (AOH), alternariol methyl ether (AME), and tenuazonic acid (TeA). These toxic secondary metabolites are secreted by pathogenic fungi such as *Alternaria* [6]. At low concentrations, these mycotoxins can disrupt the normal physiological and metabolic functions of the host and even induce apoptosis. They have genotoxic, mutagenic, cytotoxic, and other adverse effects. In addition, there is evidence of synergistic effects between AOH and AME [7]. Grapes (*Vitis vinifera* L.) are one of the most popular fruits in the world, with rich nutritional and medicinal value [8]. Apart from being eaten raw, grapes are also extensively processed to make wine, grape juice, raisins, and other products to satisfy a variety of customer demands [9,10,11]. Consequently, grapes hold significant economic worth within the worldwide fruit industry [12]. However, grapes are berries, which are susceptible to mechanical damage and pathogen infection while being picked, stored, and transported [13]. Among them, pathogens such as *Alternaria* can quickly invade through the wounds or skin of fruits [14]. Once the conditions are suitable, its mycelium will spread out and generate more conidia, which will lower quality and cause postharvest losses [15]. In order to protect the interests of consumers and the red grape fruit industry, it is therefore essential to gain a full understanding of the biological characteristics of *Alternaria*. With this understanding, effective control measures for preventing *Alternaria*-caused postharvest deterioration can be developed and put into practice.

Synthetic fungicides are widely used in pre- and postharvest production to control plant diseases. Consumer demand for organic fruits and vegetables free of pesticides and chemical residues is rising; nevertheless, long-term usage of improved fungicides can result in fungal strains, chemical residues, and environmental contamination of antifungal agents [16]. In this context, sulphur dioxide (SO_2_) fumigation has been shown to be an effective method of controlling the postharvest pathogen *A. alternata*. However, the optimal dosage of SO_2_ treatment is often close to causing bleaching damage, fruit cracking, and softening. In addition, potential residues of SO_2_ pose a risk to human health, limiting its practical use for prolonged exposure in production practices [17]. In contrast, dielectric-barrier discharge plasma (DBD) [18] and ultraviolet irradiation (UVI) [19] have developed into large-scale environmental protection and physical sterilization methods for controlling postharvest diseases. The safety profiles of these non-thermal sterilizing methods are enhanced. Unfortunately, their high cost and sophisticated nature prevent them from being used in areas with poor infrastructure. Moreover, microbial antagonists are known to be highly plastic in their growth and reproduction stages, sensitive to environmental factors, and difficult to ensure stability, thus resulting in limited therapeutic effects in practical applications. Thus, over the past ten years, more research has been conducted on biocontrol agents (BCAs) based on natural substances, paying particular emphasis to the use of plant extracts or essential oils as fungicides in place of their synthetic counterparts [20].

For example, carvacrol is a monoterpenoid phenol, a light-yellow oily liquid with lipophilicity and good inhibitory effects on various pathogenic bacteria. It has been reported to cause structural damage to cell walls and membranes [21], and it also exhibits strong antibacterial and antifungal properties in the field of food preservation. Indeed, the excellent antibacterial properties of carvacrol make it a suitable substitute for ketanol and carbolic acid in the dental field [22]. In addition, it is reported that compared to other plant essential oil components, carvacrol exhibits superior postharvest disease inhibitory activity. In a recent study by MI et al. [23], it was found that the growth of *A. alternata* was significantly inhibited after treatment with carvacrol, ultimately improving the storage quality parameters of kiwifruit. Due to the harmful effects of high concentrations of carvacrol on fruits, it is not suitable to choose high concentrations in practical applications [24]. However, little is known about the possible outcomes and workings of applying low carvacrol concentrations (≤*EC_50_*) to stop *A. alternata*-caused postharvest illnesses. Thus, the goal of this study was to investigate the antifungal potential and control effect of low concentrations of carvacrol on the main pathogenic fungus, *A. alternata*, in red grapes after harvest. In addition, we aimed to study the quality parameters and disease resistance of Streptomyces in vivo and the effects of low-dose carvacrol treatment on red grape infections.

## 2. Materials and Methods

### 2.1. Materials and Reagents

Red grape (Krenson) samples were purchased from a local supermarket in Yucheng District, Ya ’an City, Sichuan Province, during the growing season. Fruits that exhibited uniformity in color and size, consistent maturity, and no mechanical damage were chosen, and pathogens were isolated after natural onset and decay.

Potato dextrose agar medium (PDA) was purchased from Beijing Aoboxing Biotechnology Co., Ltd., Beijing, China. Wort agar medium (MEA) was purchased from Qingdao Hi-Tech Industrial Park Haibo Biotechnology Co., Ltd., Qingdao, China.

Carvacrol, thymol, geraniol, citral, L-menthol, menthone, anisaldehyde, linalool, citronellal, trans-2-hexenal, diallyl disulfide, trans-caryophyllene, piperoneone, eugenol, and anethole (Analytical pure) were all purchased from RON Reagent Shanghai Yi En Chemical Technology Co., Ltd., Shanghai, China. All other chemicals and reagents were of analytical grade and were purchased from Chengdu Cologne Chemical Co., Ltd., Chengdu, China.

### 2.2. Isolation and Identification of Pathogens

#### 2.2.1. Isolation and Purification of Pathogens

Pathogenic fungi of red grapes were injected as described by Zhang et al. [25]. The experiment was conducted on an ultraclean workbench (SW-CJ-1F, Shanghai Bangxi Instrument Technology Co., Ltd., Shanghai, China), and related materials were sterilized using a high-pressure steam sterilization pot (GI54DS, Xiamen Zhihui Instrument Co., Ltd., Xiamen, China). It was then placed in a biochemical incubator (LRH-250F, Shanghai Yiheng Technology Co., Ltd., Shanghai, China) and cultured (28 °C, 48 h). Subsequently, the edge of the colony mycelium was chosen based on the cultivation methods until one colony formed.

#### 2.2.2. Morphological Identification

Fungi were inoculated on potato dextrose agar (PDA) and cultured (28 °C, 5 d). We looked at the colony’s size, morphology, color, texture, and other aspects. Water lenses were made so that hyphae and spore morphological characteristics might be seen and photographed using a biomicroscope (N-126, Ninon Corporation, Tokyo, Japan). The species of fungi were preliminarily identified by using a fungal identification manual.

#### 2.2.3. Molecular Biology Assay

After fungi were purified, their genomic DNA was amplified using the universal primers ITS1 and ITS4 of the fungal ribosomal gene transcribed spacer. The amplified material was then transferred to the Chengdu branch of Beijing Qingke Biotechnology Co., Ltd., Beijing, China for sequencing.

#### 2.2.4. Pathogenicity of Pathogens

The purified fungi were reversely infected in accordance with Koch’s rule [26]. When disease occurred, pathogenic fungi were isolated and purified again, and compared with the previously obtained fungi.

### 2.3. Biological Characteristics of Pathogenic Fungi

The growth curves of pathogenic fungi were determined, and the effects of different growth conditions were investigated. The media included potato dextrose agar (PDA), potato sucrose agar (PSA), Chagall medium (Czapek), wort agar medium (MEA), bean sprout juice medium (BSA), oat agar medium (OA), and red grape agar medium (RGA). Carbon and nitrogen sources were also used (carbon sources: sucrose, mannitol, glucose, soluble starch, D-maltose, lactose, and D-fructose. Nitrogen sources: sodium nitrate, ammonium sulfate, ammonium nitrate, potassium chloride, potassium nitrate, L-Alanine, and urea). The temperatures were 4, 10, 16, 22, 28, 34, 40. The pH values were 3, 4, 5, 6, 7, 8, 9. The light conditions included full light, alternating light and dark, and full dark. The fatal temperatures of pathogenic fungus hyphae and spores, as well as colony diameter, spore production, and spore germination, were measured. Each treatment was repeated three times, and the average value was taken to determine the optimal growth conditions of pathogenic fungi.

#### 2.3.1. Growth Curve of Pathogenic Fungi

With a few minor adjustments, spore suspension was made as described by Zhang et al. [27]. The concentration of pathogenic fungi was adjusted with sterile water to 10^5^ spores/mL for later use.

A 200 uL spore suspension was cultured in 100 mL of PDB medium for 7 d at 28 °C and 110 rpm in a water bath thermostatic shaker (SHA-BA Shanghai Yiheng Technology Co., Ltd., Shanghai, China), with samples being obtained every 12 h. After centrifuging the samples (6000× *g* r/min for 10 min) to extract the culture solution, they were dried to a consistent weight in an electric blast drying oven (DHG-9245A Shanghai Yiheng Technology Co., Ltd., Shanghai, China) set at 60 °C. The growth curve was then drawn using the dry weight of hyphae as the ordinate and the culture time of pathogenic fungi as the abscissa.

#### 2.3.2. Culture Medium

Seven different types of culture media were infected with the pathogenic fungal cake (28 °C, 5 d). The total number of spores was counted using a blood cell counting plate, and the colony diameter was measured using the cross method.

#### 2.3.3. Carbon and Nitrogen Sources

Seven carbon and nitrogen sources were employed in place of sucrose and sodium nitrate, and the other procedures were the same as above. Czapek medium was used as the basic medium. The carbon and nitrogen sources were prepared into 1% solution with sterile water and pathogenic fungal spores were collected. A 25 μL spore suspension was coated on the concave slide and cultured (28 °C, 4 h). Three spore germination zones were observed. The spore germination rate was estimated, and the total number of spores should not be less than 150. The length of the germ tube must be greater than half of the diameter of the spore to germinate. The germination rate of spores was calculated according to the following formula:Germination rate of spores = (n_0_/n) × 100%
where n_0_ is the number of germinated spores and n is the total number of spores.

#### 2.3.4. Temperature

Other steps were the same as above, in addition to temperature.

#### 2.3.5. Lethal Temperature

The test tube holding the bacterial cake or spore solution was heated (35, 40, 45, 50, 55, 60, and 65 °C for 10 min) in a digital display thermostatic water bath (HH-4 Shanghai Bangxi Instrument Technology Co., Ltd., Shanghai, China), and then cultured (28 °C, 5 d). According to the highest temperature that fungi could grow, the temperature gradient that fungi could not grow was set at an interval of 1 °C, and then the above process was repeated.

#### 2.3.6. pH

The pH of sterile water and PDA medium was adjusted by using 1 mol/L HCl and 1 mol/L NaOH. The other operation steps were the same as above.

#### 2.3.7. Light

Other steps were the same as above, in addition to lighting options.

### 2.4. Screening of Natural Bacteriostatic Agents

Pathogenic fungi were inoculated into PDA medium that contained 15 essential oil components at a specific concentration, including carvacrol, thymol, and geraniol (Table A1). By measuring the colony diameter, the growth inhibition rate was estimated using the formula shown below. The logarithm of the concentration of the essential oil’s constituents as the abscissa and the inhibition rate as the ordinate were used to calculate the virulence regression equation and the correlation coefficient R. The virulence regression equation was then used to obtain the *EC_50_* value as follows:Mycelial growth inhibition rate = [(d_0_ − d)/(d_0_ − 5)] × 100%
where d_0_ is the control colony diameter and d is the colony diameter of the experimental group.

### 2.5. Inhibitory Effect of Essential Oil Components on Pathogenic Fungi In Vitro

The mechanism of action of carvacrol was determined using scanning electron microscopy (SEM) and transmission electron microscopy (TEM) as previously described [28], with minor modifications.

The cultured pathogenic fungi were fixed in 3% glutaraldehyde solution (4 °C, 2 h), washed three times with phosphate-buffered solution (PBS; pH 7.2, 0.1 mol/L), and dehydrated in 30%, 50%, 70%, 80%, 90%, 95%, and 100% alcohol gradient for 15 min. With conductive adhesive, the sample was attached to the sample holder, which was then placed in an ion sputter (e-1045, Hitachi High Tech Nako, Tokyo, Japan) for spraying treatment. The sample was photographed using a scanning electron microscope (jsm-it700 hr, Japan Electronics, Tokyo, Japan).

The tissue was first prefixed with 3% glutaraldehyde, followed by 1% osmium tetroxide postfixation, series acetone dehydration, extended Epon 812 infiltration, and embedding. The semithin sections were stained with methylene blue, while the ultrathin sections were cut with a diamond knife and stained with uranyl acetate and lead citrate. A JEM-1400-FLASH transmission electron microscope was used to examine the sections.

### 2.6. Inhibitory Effect of Essential Oil Components on Pathogenic Fungi In Vivo and Their Preservation Effect

To ensure uniformity in size and ripeness and the absence of mechanical damage, red grapes were cut off from the stem. Pretreatment was performed as explained by Ding et al. [29]. These sterilized red grapes were used as the negative control group (CK1), and the inoculated fruits were used as the positive control group (CK2), following spraying of the sterilized red grapes’ surfaces with *A. alternata* spore suspension. The remaining fruits were divided into two groups and sprayed with 0.5 *EC_50_* and *EC_50_* concentrations of carvacrol, marked as Q1 and Q2. Each group was prepared with fifteen boxes of red grapes, and each measurement was performed three times or more to replicate the natural shelf-life state while the fruits were kept at room temperature. The disease incidence, the degree of pericarp membrane lipid peroxidation of the peel, and the activities of defense-related enzymes of the peel were evaluated and recorded over 12 d at intervals of 3 d.

#### 2.6.1. Postharvest Disease Incidence Rate

We referred to a previous study method, with minor modifications [30]. Based on the size of the fruit decay area, it was divided into different grades as described by Ding et al. [29]. The decay rate was calculated according to the following formula:Decay rate = [(i × n_i_)/4n] × 100%
where i is the level of decay and the number of fruits at that level of decay, and n is the total number of fruits per box of red grapes.

#### 2.6.2. Postharvest Changes in Physical and Chemical Properties

Each group consisted of three red grape groups that were fixed at random. The fixed group was weighed at each sampling and the findings were recorded. The weight at 0 d was the initial weight. The change in red grape hardness was measured using a fruit hardness tester (GY-4, Yueqing Edbao Instrument Co., Ltd., Yueqing, China) based on a previous report by Xu et al. [31], with some modifications. The total soluble solids (TSS) content of red grapes was determined using a digital refractometer (LH-B55, Hangzhou Luheng Biotechnology Co., Ltd., Hangzhou, China), and the method for determining titratable acid (TA) was acid–base titration, which had been slightly modified according to a previous report [32]. The weight loss rate was calculated according to the following formula:Weight loss rate = [(m_0_ − m_t_)/m_0_] × 100%
where m_0_ is the initial weight of red grapes and m_t_ is the weight of red grapes on t d.

#### 2.6.3. Degree of Membrane Lipid Peroxidation in Fruit Peel

The method used to determine the content of malondialdehyde in fruits was based on a previous study [33], with some modifications. This method required the use of a UV-Vis spectrophotometer (UV BlueStar A, Beijing Lebertech Instrument Co., Ltd., Beijing, China). The relative conductivity of the red grapes’ peel was measured using a conductivity meter (DDS-11A, Shanghai Youyou Instrument Co., Ltd., Shanghai, China), which was determined based on a previous report by Han et al. [34], with minor modifications.

#### 2.6.4. Defense-Related Enzyme Activity

Red grapes (5 g) were collected and homogenized in ice-cold PBS (5 mL, pH 7.2, 0.1 mol/L). Subsequently, the homogenates were centrifuged (4 °C, 10,000 r/min, 15 min). The resultant supernatants were then used as enzyme extracts to measure the activities of defense-related enzymes using plant-based ELISA kits from Chengdu Kuafu Technology Co., Ltd., which included superoxide dismutase (SOD), peroxidase (POD), catalase (CAT), and phenylalanine ammonia lyase (PAL).

### 2.7. Statistical Analysis

The final results were expressed as the means ± standard deviations of three independent replicates. The phylogenetic tree was drawn using MEGA-X. Excel 2022 was used to calculate data statistics. Means were compared using one-way analysis of variance and Duncan’s multi-range test using SPSS 25. *p* < 0.05 was considered significant. The Origin 2021 software was used for statistical analyses.

## 3. Results and Discussion

### 3.1. Isolation and Identification of Pathogens

The fungal group *Alternaria*, which exhibits high adaptability and a wide range of species, is a significant cause of diseases in fruits and vegetables [35]. As shown in Figure 1, the colonies of strain *LQ* on PDA medium had an almost circular shape. The hyphae were initially greyish white and radially growing, gradually deepening to greyish green or ink green. The colonies had flat and dense villi with clean edges, developed aerial hyphae, and exhibited a growth rate of 6.01 mm/d (Figure 1a). Over time, the colonies changed color from pale yellow to blackish brown (Figure 1b). Microscopic examination revealed largely unbranched mycelia with sizes ranging from 1.8 to 4.7 µm and septa (Figure 1c). The conidial chains were upright or curved and the conidia appeared yellowish brown or brown with different shapes and sizes (11.83–42.66 µm × 5.69–16.12 µm) (Figure 1c). The conidia’s surface was smooth or drum-shaped, with 0–2 longitudinal or oblique septa and 2–5 transverse septa that were constricted at the septa. The septa in the middle part of the spore body were thicker and dark brown (Figure 1d), with most having a light brown columnar short beak. The long beak had a septum and measured 2.40–25.24 µm × 2.80–5.30 µm (Figure 1c). Based on the colony characteristics and reference to fungal identification manuals, strain *LQ* was tentatively identified as *Alternaria*.

The pathogenicity test confirmed that the red grapes in the control group showed no disease response. However, the red grapes in the experimental group, whether injured or uninjured, were all infected with the fungus and produced a significant number of spores. In addition to exhibiting mycelial development and spore germination on the uninoculated side, the red grapes in the wounded, inoculated treatment group also had disease signs, which was consistent with the natural symptoms of red grapes. The fungi were reisolated and purified and found to match the characteristics of the inoculated strain. This agreement with Koch’s postulates indicated that this fungus was the main pathogen responsible for red grape storage diseases.

It was found in this study that *A. alternata* was the predominant pathogenic fungus of red grapes, which was compatible with the findings of Ghuffar S in the Rawalpindi district of Punjab province, Pakistan [36]. Furthermore, Li Z isolated *A. alternata* from the leaves of the Amur grape *Vitis amurensis* [37]. Together, these data demonstrated that *A. alternata* was the primary pathogenic fungus of grapes, and it was possible that there were no geographical variations.

Sequencing revealed a fragment size of 550 bp in strain *LQ* (Figure A1a). With 100% coverage and homology, the ITS sequences of pathogenic fungi with a higher homology were all recognized as *Alternaria species* through further homology alignment with known sequences (Figure A1b) in the NCBI database. The phylogenetic tree is shown in Figure A1c. Sequences from different *Alternaria species* formed a single large clade, with the branch confidence scores mostly above 70% and often above 90%. *LQ* clustered closely with two *A. alternata* sequences, with high (100%) branch confidence and close genetic distance. This cluster of branches was relatively independent of those of other *Alternaria species*, indicating that *LQ* could be confidently identified as *A. alternata*.

### 3.2. Biological Characteristics of LQ

The mycelium of this strain, as depicted in Figure A2f, grew slowly throughout the course of 0–24 h without significant change in weight, suggesting a lag phase. Rapid mycelial growth then transpired between 24 and 72 h, signifying a logarithmic growth phase. During 72–108 h, a little rise in mycelial weight was noted. Mycelial weight did not significantly alter between 108 and 132 h, suggesting that the strain had stabilized. The strain’s weight steadily dropped starting at 132 h, signifying the start of a decline phase.

Significant differences (*p* < 0.05) were observed in mycelial growth and spore production of strain *LQ* under different nutrient and environmental conditions, as shown in Figure 2. *LQ* showed the highest growth rate in BSA medium (60.29 ± 0.48 mm). D-maltose and ammonium nitrate supported superior growth compared to other groups. Similarly, potassium nitrate and L-alanine also promoted accelerated growth with no significant difference between them. Optimal growth occurred at 28 °C, while growth slowed significantly above 34 °C and was absent at 4 °C and 40 °C. *LQ* preferred neutral or slightly acidic conditions with pH of 6 for maximum growth, while alkaline conditions inhibited it. Mycelial growth rate was higher in the light/dark alternation and full-light treatments compared to full darkness, with no significant difference between them. Thus, the ideal conditions for *LQ* mycelial growth were BSA medium, D-maltose, ammonium nitrate, 28 °C, pH 6, and exposure to light, and the optimum conditions for sporulation were consistent with this (Figure A2). The hyphae and spores of *LQ* were effectively sterilized at 56 °C for 10 min (Table A2).

*LQ* did not produce spores in Czapek and MEA media. Of the media tested, OA was the most favorable for spore production. When mannitol and urea were used as carbon and nitrogen sources, *LQ* showed the highest spore production. However, *LQ* did not produce spores in the medium containing ammonium sulphate and L-alanine as nitrogen sources. The temperature was correlated with spore production, which peaked at 34 °C. At 4 and 40 °C, no spores were generated. In contrast to mycelial growth, alkaline and dark conditions favored *LQ* spore production. At pH 9, spore production reached (2.117 ± 0.0764) × 10^6^ spores/mL. Therefore, OA medium, mannitol, urea, 34 °C, pH 9, and dark conditions were recommended for optimal *LQ* spore production.

Remarkably, we discovered that there was no positive correlation between the strain’s capacity for mycelial development and sporulation. The specific results were different from Wang et al. [38], which may be because different choices were made for culture medium types and environmental factors. Due to the differences in the utilization of nitrogen sources by *LQ*, it may be safer to choose ammonium sulfate as a nitrogen fertilizer compared to nitrate and urea fertilizers.

### 3.3. Effect of Carvacrol on LQ In Vitro

The growth of *LQ* was inhibited to varying degrees by 15 essential oil ingredients, including carvacrol, thymol, and geraniol, with the inhibition proportional to the concentration (Figure A3). As shown in Table A3, the toxicity regression equations for these components showed good linear correlation (R > 0.97). Carvacrol exhibited the strongest antibacterial effect, with an *EC_50_* of 36.71 µg/mL, surpassing the other 14 ingredients. Thymol, geraniol, citral, and L-menthol followed, with *EC_50_* values ranging from 40 to 90 µg/mL. Consequently, carvacrol was selected for subsequent experiments as an inhibitor against *LQ*. Carvacrol is a lipophilic, light-yellow oily liquid monoterpene phenol that has good pathogen inhibitory activity. It is present in a variety of oregano plants, including sage, thyme, and oregano, of which it is most prevalent in oregano and the primary active component of oregano essential oil [39]. As a compound with a variety of biological activities, it has broad application prospects. The World Health Organization (WHO) reported that when the residue of carvacrol in food was less than 50.0 mg/kg, it had no harm to human health and could be used for postharvest preservation of fruits [40].

Damage to the hyphae and spores of *LQ* increased with carvacrol concentration (Figure 3a). The control group showed regular, plump, and smooth mycelia with intact cell walls and germinated spores. In contrast, carvacrol-treated mycelia and spores showed shrinkage, which became more pronounced with increasing concentration. At *EC_50_*, wrinkles and craters similar to meteorite craters appeared and small cracks appeared on the mycelial surface. Above *EC_50_*, hyphae and spores were completely destroyed and spores did not germinate. Specifically, at 2 *EC_50_*, hyphae shrank and bent, and the spore surface suffered extensive damage with disrupted cell walls. As a result, higher carvacrol concentrations harmed *LQ* hyphae and spores more severely and prevented them from growing normally.

The TEM observations (Figure 3b) were highly consistent with the SEM results. The control group showed intact cell walls and membranes, uniformly distributed plasma membranes, abundant cytoplasm and contents, and visible organelle structures. In the 0.5 *EC_50_* and *EC_50_* treatment groups, *LQ* showed cell swelling, distorted mitochondria, and increased lipid droplets and autophagosomes, while the cell walls and membranes of pathogenic fungi remained largely intact. Significant damage to the cell wall and membrane, the loss of cell contents, a decrease in the number of lipid droplets, an unorganized internal matrix, and a distortion in the shape of organelles were the outcomes of concentrations higher than *EC_50_*. It was hypothesized that at lower concentrations (≤*EC_50_*), carvacrol inhibited *LQ* growth by causing damage to mitochondria and causing aberrant formation of lipid droplets, while at higher concentrations (>*EC_50_*), carvacrol was likely to cause irreparable cell damage by rupturing the integrity of the *LQ* cell wall and membrane.

Through in vitro experiments, we found that carvacrol could inhibit the growth of mycelium and spore germination of *A. alternata*, and there was a certain dose relationship. On the one hand, carvacrol inhibited important enzyme activities linked to energy metabolism in pathogenic fungal cells, which might have further harmed mitochondria by altering the mitochondrial shape and causing an aberrant increase in lipid droplets. Insufficient energy supply to the cells could lead to the activation of autophagosomes, which are important molecules for maintaining cell homeostasis [41]. On the other hand, carvacrol might inhibit important enzymes involved in the synthesis of cell walls and membranes, which hinders the production of substances like chitin and ergosterol. Additionally, accumulated ROS resulted in membrane lipid peroxidation, which damaged the integrity of cell walls and membranes and caused pathogenic fungal cells to cease growing and die [42].

### 3.4. Effect of Carvacrol on the Physical and Chemical Properties of LQ-Inoculated Red Grapes

According to the results shown in Figure 4a, carvacrol effectively inhibited fruit decay caused by *LQ* at different concentrations. The positive control group (CK2) showed a significant increase in decay rate from day 0, which was significantly higher than the other treatment groups (*p* < 0.05). On 12 d, the degradation rate reached 63.37 ± 1.75%. The degradation rate in the CK1, Q1, and Q2 groups significantly increased starting on 6 d, and by 9 d, notable changes were seen. On 12 d, the degradation rates were 11.56 ± 0.89%, 29.29 ± 1.29%, and 14.73 ± 0.69%, respectively. The Q2 group showed stronger inhibitory effects and closely resembled the CK1 group, with a significant difference observed only on 12 d (Figure 4f).

As shown in Figure 4b, carvacrol effectively slowed the weight loss caused by *LQ*, in line with the degradation trend. With the highest rate of 12.27 ± 0.38%, CK2 showed increased water loss and faster respiration as a result of *LQ*. On 9 d, both essential oil treatment groups and CK1 showed similar patterns with no significant differences (*p* > 0.05). In general, carvacrol slowed fruit respiration and reduced water loss, with Q2 giving the most favorable results.

As shown in Figure 4c, carvacrol considerably inhibited the deterioration of red grapes and preserved their hardness during storage, while *LQ* caused the grapes to soften more during storage. CK2 showed the most severe softening and the most rapid decrease in hardness. Fruits treated with carvacrol showed a milder decrease. On 12 d, the hardness of the four groups was 3.71 ± 0.07, 0.6 ± 0.13, 3.12 ± 0.09, and 3.68 ± 0.09 N, respectively. Compared to 0 d, the hardness decreased by 26.97%, 87.93%, 37.35%, and 25.35%, respectively. Q2 and CK1 showed similar trends without significant differences until the last day (*p* > 0.05). In conclusion, carvacrol reduced fruit softening and retained firmness, with Q2 showing greater efficacy than Q1.

TSS in all four treatment groups had a similar trend, rising at first and then dropping over time (Figure 4d). This is most likely because TSS functions as a structural carbohydrate that is mostly hydrolyzed during fruit metabolism [43]. CK2 showed a sharp increase to 20.2 ± 0.02%, which decreased after the third day. CK1 and Q1 reached their peak on 6 d with contents of 18.61 ± 0.04% and 18.16 ± 0.08%, respectively. Notably, Q2 reached its peak on 9 d (18.89 ± 0.04%). On 12 d, the TSS of Q2 was significantly higher than that of CK1, indicating the beneficial effect of higher carvacrol concentration on red grape preservation. Overall, carvacrol inhibited the metabolism of *LQ*-infected red grapes, with Q2 showing the best results.

TA of red grapes dropped during storage (Figure 4e), most likely as a result of additional materials and organic acids being consumed during fruit respiration. In the later stages of storage, CK2 had a significantly lower TA than the other three groups (*p* < 0.05), while the essential oil-treated group showed a mitigated decline. On 12 d, the TA of CK2 was 0.23 ± 0.02%, a decrease of 74.22% compared to 0 d, while the other three groups (CK1/Q1/Q2) showed decreases of 58.29%, 67.29%, and 58.52%, respectively. Overall, this suggests that carvacrol treatment could delay the rate of titratable acid consumption, with Q2 showing a slower metabolic rate in red grapes infected with *LQ*.

Red grapes naturally have life, and during storage, they engage in a variety of biological processes like transpiration and respiration, which cause the fruit to soften after dehydration. Pectin hydrolysis and depolymerization immediately contribute to cell damage and cell wall rupture, which in turn cause fruit softening and hardness loss [44]. It has been reported that carvacrol-containing biofilms effectively preserve tomatoes and persimmons. After twenty days, tomatoes’ inhibition rate reached 90%, while by two months, persimmons showed no damaged fruit [45]. In addition, carvacrol could significantly maintain the hardness of fresh-cut apples, prevent their softening, and significantly alleviate the decrease in TSS and TA contents [46]. This was consistent with the research results. All these indicated that carvacrol had broad-spectrum antibacterial activity, which could reduce the aging of fruits, slow down the deterioration of quality, prolong the storage time of fruits, and produce effective fresh-keeping effects. However, the preservation effect may vary with the type, processing method, and concentration of fruits and vegetables.

### 3.5. Effect of Carvacrol on Membrane Lipid Peroxidation of LQ-Inoculated Red Grapes

There were variations in the degree of membrane lipid peroxidation of red grape peel in each group over the course of storage (Figure 5). It was observed that the change trend of MDA and relative conductivity increased. Additionally, the CK2 group had a significantly higher degree of membrane lipid peroxidation than the other three groups (*p* < 0.05). The final byproduct of membrane lipid peroxidation is the content of MDA, which is widely used to assess the level of senescence damage to fruit cells in conjunction with relative conductivity [34]. Thus, this case showed that when red grapes were infected with *LQ*, the degree of membrane lipid peroxidation rose, resulting in ongoing accumulation of MDA, damage to the cell membrane, electrolyte leakage, and concurrently elevated relative conductivity. On 12 d, the MDA content of CK2 reached 19.72 ± 0.41 μmol/g, while Q1 and Q2 had a lower concentration with a significant difference (*p* < 0.05). According to this study, carvacrol may be able to stop red grape peel’s membrane lipids from peroxiding. Overall, carvacrol can effectively delay fruit senescence, preserve the integrity of cell membrane, and prevent oxidative damage to pericarp cell membrane lipids caused by an infection with *LQ*, with Q2 showing greater efficacy.

Active oxygen buildup in red grapes during storage caused the membrane system to be attacked and the membrane structure to be degraded, which increased the fruit’s MDA level, caused electrolyte leakage, and accelerated fruit aging. The reason why carvacrol could alleviate this situation may be because it had an inhibitory effect on *A. alternata*, which initially inactivated some pathogenic fungi and prevented their infection and colonization. This was consistent with the results of Chen et al. [47].

### 3.6. Effect of Carvacrol on Defense-Related Enzyme Activities of LQ-Inoculated Red Grapes

All defense enzymes’ activity increased during storage at first before declining once more (Figure 6). The peak value for the CK2 group was higher than the CK1 group, occurred 3–6 d earlier, and then dropped off quickly. The enzyme activities decreased in the CK1 group on 12 d. When carvacrol was introduced, it interacted with *LQ* and changed the defense enzymes’ activity in a statistically significant way (*p* < 0.05). The overall enzyme activities of the Q1 and Q2 groups were significantly better than those of the CK1 group and persisted at a higher level until the last day. Particularly, the enzyme activities of the Q2 and CK1 groups peaked after 9 d of storage, while SOD, POD, CAT, and PAL activities were 1.25, 1.14, 1.25, and 1.2 times those of CK1, respectively. In comparison to the CK2 group, the peak times of the CAT and PAL enzyme activities in the Q1 group were delayed by 3 d, while the Q2 group saw a 9 d delay in the peak time of each enzyme activity. Overall, carvacrol increased disease resistance, stimulated the activities of antioxidant defense enzymes, and postponed the aging of red grapes, with Q2 demonstrating greater preservation impact.

ROS are highly active and toxic, which could cause damage to lipids, proteins, DNA, and other substances, which would result in oxidative stress. SOD could convert O^2−^ to H_2_O_2_, and excessive H_2_O_2_ could promote the peroxidation of membrane lipids. Two ROS scavengers, POD and CAT, work together to get rid of them and protect plant cells [48]. This study showed that carvacrol raised the SOD, POD, CAT, and PAL activities in red grapes during storage. Carvacrol is a naturally occurring monoterpene phenol that includes both hydrophobic benzene rings and hydrophilic phenolic hydroxyl groups, which may be the reason for this. It may, therefore, readily pass through epidermal cells that comprise the cell wall, boost the production of ROS, change the membrane’s permeability, stimulate an antioxidant response, and strengthen the disease resistance of red grapes. In summary, carvacrol has a significant inhibitory effect on *A. alternata* [49].

## 4. Conclusions

*A. alternata* has strong pathogenicity under growth conditions that are not harsh and strong vitality. It has disease effects on red grapes and affects the quality and economic benefits of red grapes. In this study, it was discovered that carvacrol exhibited a broad-spectrum antifungal action in vitro against *A. alternata*, the main postharvest fungal pathogen of red grapes. More specifically, upon treatment with carvacrol at a low concentration (<*EC_50_*), fungal hyphae and spores began to distort. At *EC_50_*, the complete mitochondrial morphology of *A. alternata* was destroyed, and lipid droplets were abnormal, which led to the disorder of cell energy metabolism and ultimately affected the normal growth of this strain. However, when exposed to high concentrations of carvacrol (>*EC_50_*), spores shrank or even broke, hyphae were completely deformed, the cell wall and membrane integrity was lost, and the cytoplasm was exhausted, which resulted in cell death and growth restriction. At the appropriate concentration of carvacrol (0.5 *EC_50_*/*EC_50_*), red grapes retained a higher firmness and TSS, TA, and other quality parameters. Overall, carvacrol is capable of preventing *Alternaria alternata* from growing, and it could be utilized as a control measure for preventing *Alternaria alternata*-caused red grape decay diseases.

## Figures and Tables

**Figure 1 foods-12-04305-f001:**
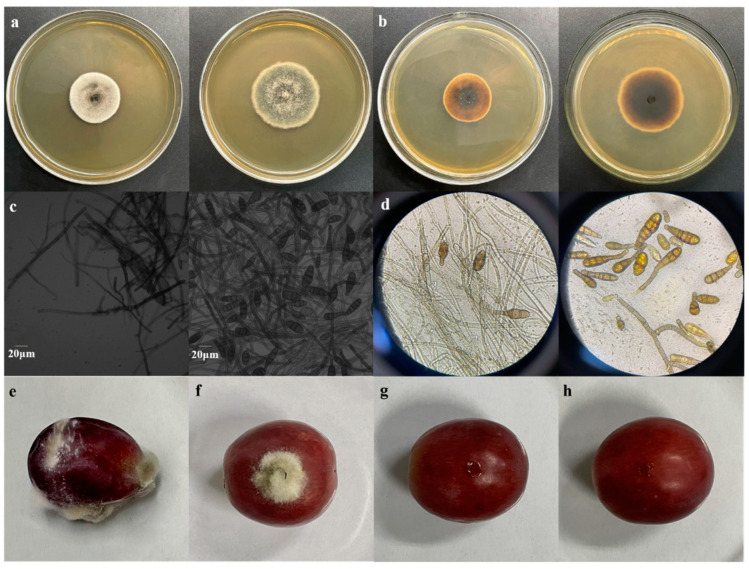
Growth and morphological characteristics of *LQ* and diseases of red grapes caused by reverse inoculation: (**a**) colony in PDA medium (front); (**b**) colony in PDA medium (reverse); (**c**) microscopic images of hyphae and spores (40×); (**d**) observation of hyphal and spore morphology; (**e**) inoculation of pathogenic fungi in injured fruits; (**f**) inoculation of pathogenic fungi without injury; (**g**) injured control; and (**h**) no-injury control.

**Figure 2 foods-12-04305-f002:**
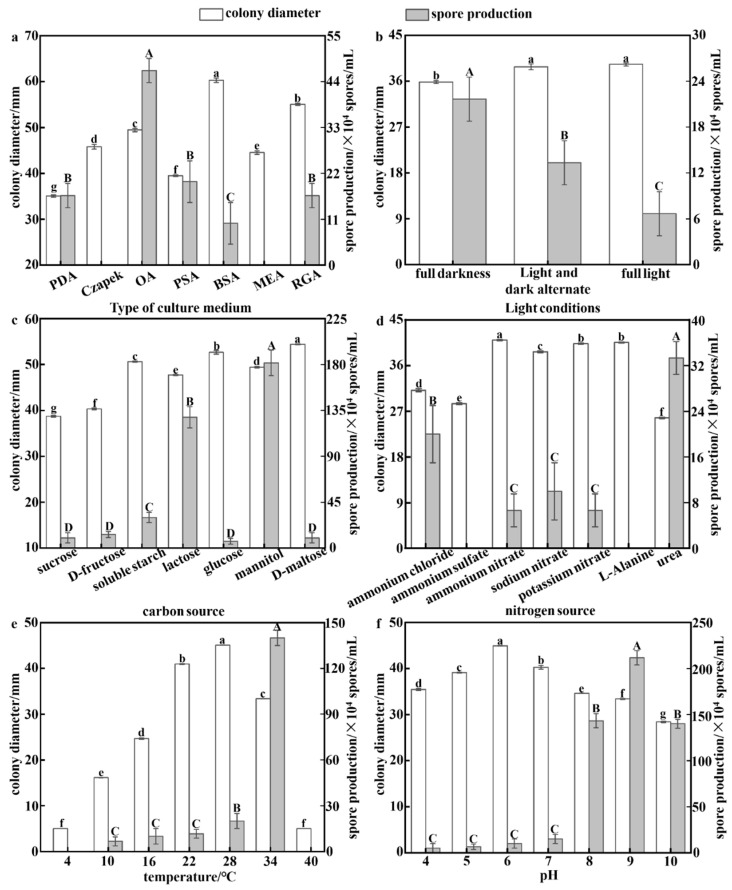
Colony diameter and spore production of *LQ* under different growth conditions: (**a**) culture media; (**b**) carbon sources; (**c**) nitrogen sources; (**d**) temperature; (**e**) pH; and (**f**) lighting conditions. Different minuscule letters indicate significant differences in colony diameter (*p* < 0.05). Different capital letters indicate significant differences in spore production (*p* < 0.05).

**Figure 3 foods-12-04305-f003:**
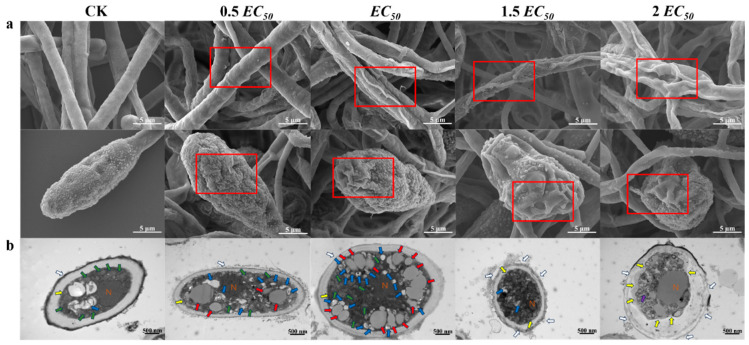
SEM and TEM results: (**a**) surface morphology of hyphae and spores observed under SEM, where the red box indicates the twisted morphology of hyphae and the wrinkled and damaged state of spores; (**b**) structure of cells observed under TEM, where the white arrows indicate cell walls, the yellow arrows show cell membranes, the red arrows show mitochondria, the green arrows show lipid droplets, the blue arrows show autophagosomes, the purple arrows indicate the loss of cytoplasmic material, and N indicates the cell nucleus.

**Figure 4 foods-12-04305-f004:**
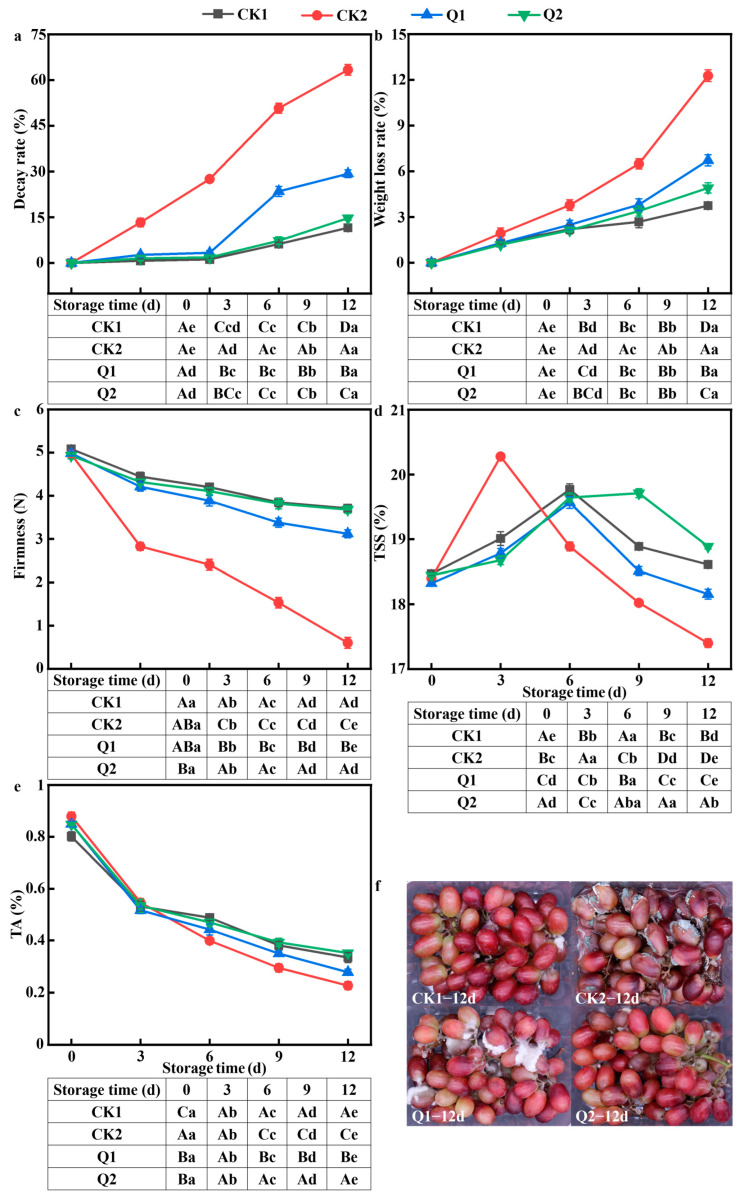
Effect of carvacrol on the quality of red grapes during storage with different treatments: (**a**) decay rate; (**b**) weight loss rate; (**c**) hardness; (**d**) TSS; (**e**) TA; and (**f**) decay on the last day of storage. Different minuscule letters indicate significant differences in colony diameter (*p* < 0.05). Different capital letters indicate significant differences in spore production (*p* < 0.05).

**Figure 5 foods-12-04305-f005:**
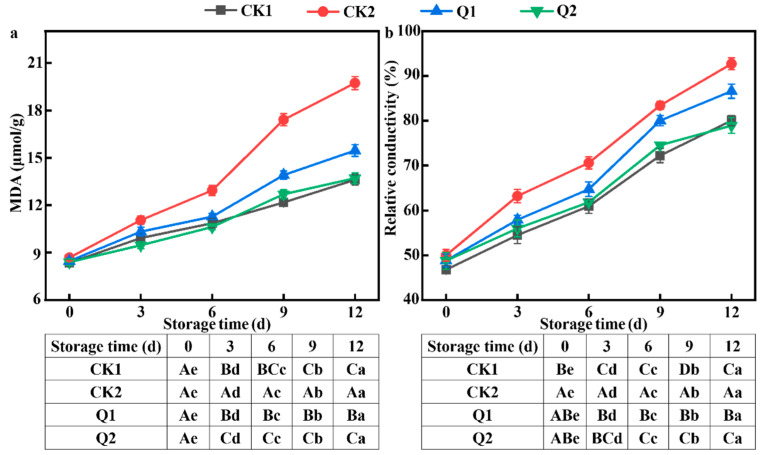
Effects of carvacrol on the quality of *LQ*-inoculated red grapes during storage: (**a**) MDA and (**b**) relative conductivity. Different minuscule letters indicate significant differences in colony diameter (*p* < 0.05). Different capital letters indicate significant differences in spore production (*p* < 0.05).

**Figure 6 foods-12-04305-f006:**
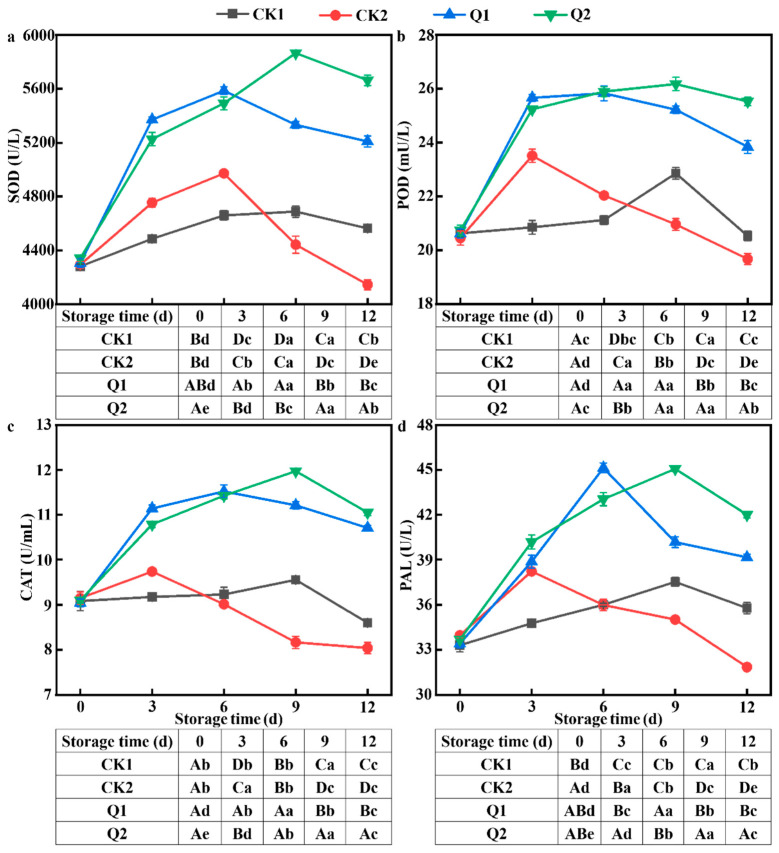
Effect of carvacrol on defense-related enzyme activities of red grapes during storage with different treatments: (**a**) SOD; (**b**) POD; (**c**) CAT; and (**d**) PAL. Different minuscule letters indicate significant differences in colony diameter (*p* < 0.05). Different capital letters indicate significant differences in spore production (*p* < 0.05).

## Data Availability

The data presented in this study are available on request from the corresponding author.

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
