# Peer review of "Carvacrol Treatment Reduces Decay and Maintains the Postharvest Quality of Red Grape Fruits (Vitis vinifera L.) Inoculated with Alternaria alternata"

_foods, 2023, doi:10.3390/foods12234305_

Round 1

Reviewer 1 Report

Comments and Suggestions for Authors

Dear author,

Thanks for your good report. My comments and questions are in the attached file.

Regards

Comments on the Quality of English Language

It is acceptable.

Author Response

Point 1: Abstract needs rewrite. Methods should be explained more. Abstract is also too short.
Response 1: Revised. Thanks, your opinion is very pertinent.

Point 2: To modify the total soluble solid to the total soluble solids.
Response 2: Revised. Thanks, your opinion is very pertinent.

Point 3: Please add a paragraph to introduction and show why you select red grape? What is problems about red grapes during storage and what previous studies have done in this regard using related recent papers.
Response 3: Revised. Thanks, your opinion is very pertinent.

Point 4: “it was found that the growth of A. alternata was significantly improved after treatment with carvacrol, ultimately improving the storage quality parameters of kiwifruit.” Decreased?
Response 4: Revised. Here should be “inhibited”. Thanks, your opinion is very pertinent.

Point 5: “Due to the harmful effects of high concentrations of carvacrol on fruits, it is not suitable to choose high concentrations in practical applications.” Reference? What is allowed amount?
Response 5: Thank you very much for your review. Reference has been added in this report. In addition, the World Health Organization has shown that carvacrol residues in food are not more than 50.0 mg/kg, which is not harmful to the health of consumers and can be used for post-harvest preservation of fruits.

Point 6: Low concentrations of parsley (≤ EC50)???
Response 6: Revised. Here should be “the low concentrations of carvacrol (≤ EC50)”. Thanks, your opinion is very pertinent.

Point 7: “In addition, we aim to study the quality parameters and disease resistance of Streptomyces in vivo and low dose parsley phenol treated red grapes infections.” Is it in this report? If not, remove from introduction.

Response 7: Thank you very much for your review. This is described in this report.

Point 8: “Materials and Methods” is it in this report? If not, remove from introduction.

Response 8: Thank you very much for your review. This is described in this report.

Point 9: “mycelial growth inhibition rate” and “decay rate”, M and D capital.
Response 9: Revised. Thanks, your opinion is very pertinent.

Point 10: “following spraying of the sterilized blueberry surfaces with A. alternata spore suspension on the sterilized fruits surfaces.” What is blueberry here?
Response 10: Revised. Here should be “red grapes”. Thanks, your opinion is very pertinent.

Point 11: It has been reported that the biofilm containing carvacrol had good preservation effect on tomatoes and persimmons. Please add references.
Response 11: Revised. Thanks, your opinion is very pertinent.

Point 12: Did you check fruit taste and sensorial quality too? Essential oils have negative impact on fruit taste. So, I think you can not recommend carvacrol for fruits and vegetables (edible fruits with skin) preservation without doing sensorial experiments. So, please rewrite conclusion.

Response 12: Revised. Thanks, your opinion is very pertinent.

Reviewer 2 Report

Comments and Suggestions for Authors

extensive reading by English-native reader is necessary for a suitable publication in this international journal

Added good recommendation in abstract 

Also write your objective clear 

how many replicate you used in your experiment 

it is not clear for me how you inoculated the plants ?

the section of isolation totally unclear

Author Response

Point 1: Added good recommendation in abstract.
Response 1: Revised. Thanks, your opinion is very pertinent.

Point 2: Also write your objective clear.
Response 2: Revised. Thanks, your opinion is very pertinent.

Point 3: How many replicates you used in your experiment.
Response 3: Thank you very much for your review. In this experiment, a negative control group (CK1), a positive control group (CK2) and two experimental groups with different concentrations (Q1/Q2) were set up. Each group was loaded with 20 boxes of red grapes about 130 g.

Point 4: It is not clear for me how you inoculated the plants? The section of isolation totally unclear.
Response 4: Thank you very much for your review. In this experiment, disease-free red grapes were soaked in a 2% sodium hypochlorite solution for 5 min, washed with sterile water, and dried at 25℃. These sterilized red grapes were used as the negative control group (CK1). Following spraying of the sterilized red grapes surfaces with 2.5 mg/g of a 1×105 CFU/mL A. alternata spore suspension, and the inoculated fruits were used as the positive control group (CK2). The remaining fruits were divided into two groups and sprayed with 0.5 EC50, EC50 concentration of carvacrol, marked as Q1 and Q2.

Reviewer 3 Report

Comments and Suggestions for Authors

_ You should mention the organ used Red grapes Fruits as also leaves are used as food & can be infested

Comments on the Quality of English Language

- Minor editing are needed as mentioned in my comments in the attached file

Author Response

Point 1: To modify the red grapes in the title to red grapes fruits.
Response 1: Revised. Thanks, your opinion is very pertinent.

Point 2: To modify by MI to by MI et al.
Response 2: Revised. Thanks, your opinion is very pertinent.

Point 3: Weight loss rate = [(m0 - mt) / m0] × 100%

where m0 is the initial weight of red grapes and mt is the weight of red grapes on t d. t d ????? meaning.
Response 3: Thank you very much for your review. Here t represents the number of days for each sampling, such as 3d, 6d, 9d, and 12d.

Point 4: H2O2 " Capital O".
Response 4: Revised. Thanks, your opinion is very pertinent.

Round 2

Reviewer 1 Report

Comments and Suggestions for Authors

ِDear author,

Thanks for the revision file.

Please use Italic for Vitis vinifera in the title.

Also, use italic for it again in line 54.

Line 63: financial losses should be postharvest losses.

Comments on the Quality of English Language

It is OK.

Author Response

Point 1: Please use Italic for Vitis vinifera in the title and use italic for it again in line 54.

Response 1: Revised. Thanks, your opinion is very pertinent.

Point 2: Line 63: financial losses should be postharvest losses.

Response 2: Revised. Thanks, your opinion is very pertinent.

Reviewer 2 Report

Comments and Suggestions for Authors

extensive reading by English-native reader is necessary for a suitable publication in this international journal

Comments on the Quality of English Language

extensive reading by English-native reader is necessary for a suitable publication in this international journal

Author Response

Point 1: Extensive reading by English-native reader is necessary for a suitable publication in this international journal.

Response 1: Revised. Thanks, your opinion is very pertinent.